# TIM-3 Expression and M2 Polarization of Macrophages in the TGFβ-Activated Tumor Microenvironment in Colorectal Cancer

**DOI:** 10.3390/cancers15204943

**Published:** 2023-10-11

**Authors:** Masanori Katagata, Hirokazu Okayama, Shotaro Nakajima, Katsuharu Saito, Takahiro Sato, Mei Sakuma, Satoshi Fukai, Eisei Endo, Wataru Sakamoto, Motonobu Saito, Zenichiro Saze, Tomoyuki Momma, Kosaku Mimura, Koji Kono

**Affiliations:** 1Department of Gastrointestinal Tract Surgery, Fukushima Medical University School of Medicine, Fukushima 960-1295, Japan; m-ktgt@fmu.ac.jp (M.K.); shotaro@fmu.ac.jp (S.N.); k-yamame@fmu.ac.jp (K.S.); taka4269@fmu.ac.jp (T.S.); msakuma@fmu.ac.jp (M.S.); sfukai@fmu.ac.jp (S.F.); eisei-e@fmu.ac.jp (E.E.); ws1024@fmu.ac.jp (W.S.); moto@fmu.ac.jp (M.S.); z-saze@fmu.ac.jp (Z.S.); tmomma@fmu.ac.jp (T.M.); kmimura@fmu.ac.jp (K.M.); kojikono@fmu.ac.jp (K.K.); 2Department of Multidisciplinary Treatment of Cancer and Regional Medical Support, Fukushima Medical University School of Medicine, Fukushima 960-1295, Japan; 3Department of Blood Transfusion and Transplantation Immunology, Fukushima Medical University School of Medicine, Fukushima 960-1295, Japan

**Keywords:** colorectal cancer, immunotherapy, TGFβ, TIM-3, M2 macrophage

## Abstract

**Simple Summary:**

TGFβ and TIM-3 have been implicated in driving immune evasion and poor response to anti-PD-1/PD-L1 immune checkpoint inhibitors (ICIs). Using large-scale transcriptomic data analysis, followed by immunohistochemistry and in vitro experiments, we demonstrated a strong link between TIM-3 and M2-like polarization of macrophages in the TGFβ-rich tumor microenvironment in colorectal cancer (CRC). We found that the transcriptional signatures of TIM-3, M2 macrophage and TGFβ-responsive stromal activation were specifically derived from the stromal components and were correlated with each other, and that they were robustly upregulated in CMS4, CMS1 and MSI CRCs. Our study suggests that the blockade of TGFβ or TIM-3, combined with ICIs, may be a promising strategy to circumvent the resistance to current ICIs for CMS4, CMS1 and MSI CRCs, where TIM-3 is highly expressed along with increased TGFβ.

**Abstract:**

TGFβ signaling in the tumor microenvironment (TME) drives immune evasion and is a negative predictor of immune checkpoint inhibitor (ICI) efficacy in colorectal cancer (CRC). TIM-3, an inhibitory receptor implicated in anti-tumor immune responses and ICI resistance, has emerged as an immunotherapeutic target. This study investigated TIM-3, M2 macrophages and the TGFβ-activated TME, in association with microsatellite instability (MSI) status and consensus molecular subtypes (CMSs). Transcriptomic cohorts of CRC tissues, organoids and xenografts were examined (*n* = 2240). TIM-3 and a TGFβ-inducible stromal protein, VCAN, were evaluated in CRC specimens using immunohistochemistry (*n* = 45). TIM-3 expression on monocytes and generated M2 macrophages was examined by flow cytometry. We found that the expression of *HAVCR2* (TIM-3) significantly correlated with the transcriptional signatures of TGFβ, TGFβ-dependent stromal activation and M2 macrophage, each of which were co-upregulated in CMS4, CMS1 and MSI CRCs across all datasets. Tumor-infiltrating TIM-3^+^ immune cells accumulated in TGFβ-responsive cancer stroma. TIM-3 was increased on M2-polarized macrophages, and on monocytes in response to TGFβ treatment. In conclusion, we identified a close association between TIM-3 and M2-like polarization of macrophages in the TGFβ-rich TME. Our findings provide new insights into personalized immunotherapeutic strategies based on the TME for CRCs.

## 1. Introduction

Colorectal cancer (CRC) is the most common gastrointestinal cancer and the third leading cause of cancer-related deaths worldwide [1]. CRC is a heterogeneous disease, with different sets of genetic events associated with diverse clinical characteristics and drug sensitivity. During the past decade, the genomic landscape of CRC has been well-established, and treatment strategies have been personalized according to genomic alterations, including *RAS* and *BRAF* mutations and microsatellite instability (MSI); however, specific therapies are only available for a small proportion of patients [2,3]. Numerous studies have recently shown that the tumor microenvironment (TME), in which cancer cells are surrounded by cancer-associated fibroblasts (CAFs), T-cells, tumor-associated macrophages (TAMs), many other cells and their extracellular matrix (ECM) proteins, may be the key driver of clinical and therapeutic outcomes in CRC [4,5,6]. In 2015, the international consortium identified four consensus molecular subtypes (CMSs) based on the gene expression profiles of bulk tumors that encompass mixed signals derived from both cancer cells and stromal cells [7]. CMS1 is closely associated with MSI, mutant *BRAF* and immune infiltration. CMS2 shows canonical WNT and MYC signaling activation. CMS3 is enriched for *RAS* mutations and metabolic dysregulation. CMS4 is defined as the mesenchymal subtype that displays prominent transforming growth factor-β (TGFβ) activation, stromal infiltration, angiogenesis and an immunosuppressive TME [2,7]. The CMS classification not only provides valuable insight into tumor biology and therapeutic strategies, but also stratifies clinical outcomes. Patients with CMS4 tumors exhibited the poorest survival rates. In this context, studies have focused more on the role of TGFβ in shaping the tumor-promoting TME in CRC, supporting the potential of TGFβ inhibition in cancer immunotherapy [4,6,8,9,10].

Dysfunctional exhausted T-cell subsets have been shown to express multiple co-inhibitory or checkpoint receptors, such as programmed cell death-1 (PD-1), cytotoxic T-lymphocyte-associated protein 4 (CTLA-4), the T-cell immunoglobulin domain and mucin-domain 3 (TIM-3), lymphocyte-activation gene 3 (LAG-3) and T-cell immunoglobulin and the ITIM domain (TIGIT) [11]. Cancer immunotherapy with immune checkpoint inhibitors (ICIs), particularly antibodies against PD-1 and its ligand programmed death ligand 1 (PD-L1), as well as CTLA-4, has revolutionized the therapeutic management of several solid cancers, such as lung cancer and melanoma. These ICIs have also been applied to CRC treatment; however, they are only effective for 4–5% of CRC exhibiting MSI [2,12,13]. MSI tumors are likely to be immunogenic; mismatch-repair deficiency (dMMR) causes hypermutation and tumor neoantigens, contributing to the creation of the “immune hot” TME. In several clinical trials, anti-PD-1 antibodies, with or without CTLA-4 blockade, demonstrated dramatic and durable responses with prolonged survival in patients with metastatic MSI CRC, although approximately 50% of them had primary resistance to ICIs [12,13]. On the other hand, microsatellite stable (MSS) CRCs showed limited or no clinical responses to ICIs, mostly because of this type being an “immune cold” tumor [13,14]. The distinct responses to ICIs among MSI and MSS CRCs suggest that ICI resistance may be due to an immunosuppressive TME driven by several molecular programs. To this end, Tauriello et al. used a mouse model of MSS CRC displaying the CMS4 phenotype and TGFβ-activated stroma, demonstrating that increased TGFβ in the TME represented a primary mechanism of immune evasion and that TGFβ inhibitors rendered tumors susceptible to PD-1/PD-L1 blockade [9]. In our previous work, we analyzed multiple patient cohorts and identified a CMS4-like subset within MSI CRC, demonstrating the TGFβ-rich TME along with a high ICI resistance signature, suggesting that strategies of TGFβ inhibition combined with ICIs may have potential therapeutic value for some MSI CRCs [15]. Indeed, TGFβ signaling has emerged as a central mediator of immune evasion and ICI resistance across a wide range of cancers, and many clinical trials are currently testing the combination of TGFβ and PD-1/PD-L1 immune checkpoint inhibition in patients with solid cancers [10,16,17,18,19,20].

TIM-3, encoded by the *HAVCR2* gene, has recently emerged as an immunotherapy target, as it is often co-expressed alongside PD-1 on the most dysfunctional and exhausted population of tumor-infiltrating T-cells [21,22]. Moreover, TIM-3 may have a role in adaptive resistance mechanisms to ICIs, as it is upregulated in response to anti-PD-1 therapy, and the co-blockade of TIM-3 and PD-1 can overcome resistance to PD-1 therapy in preclinical models [22,23]. Furthermore, the expression of TIM-3 can serve as a marker of poor prognosis in some cancer types, including CRC [21,24,25]. Although TIM-3 was initially thought to be a sign of dysfunctional T-cells, it has been demonstrated that TIM-3 is highly expressed on several types of immune cells, particularly myeloid cells, and that myeloid cells might play a predominant role in TIM-3-mediated anti-tumor immunity [21,26,27,28,29]. However, the mechanisms underlying TIM-3 regulation in non-T-cell populations remain largely undetermined [30]. We recently demonstrated that MSI CRCs exhibiting the TGFβ-rich TME were not only characterized by ICI resistance and poor survival but also the upregulation of the M2 macrophage signature along with *HAVCR2* (TIM-3) expression [15]. In this study, we analyzed the relationship between *HAVCR2*, M2 macrophages, TGFβ-activated stroma and molecular subtypes in multiple transcriptomic and immunohistochemistry cohorts of CRCs and investigated the expression of TIM-3 in M2-polarized macrophages and TGFβ-stimulated monocytes in vitro.

## 2. Materials and Methods

### 2.1. Microarray and RNA-Seq Data Analysis, Gene Signatures

The transcriptomic datasets used in this study are summarized in Appendix A. We used the preprocessed values obtained from each dataset. If a gene was represented by multiple probes, only the probe with the highest mean expression was used. We used the following six datasets of CRC patient samples, for which CMS classification was obtained from the CRCSC synapse data portal accessed on 3 February 2022 (https://www.synapse.org/#!Synapse:syn2623706/files/) [7]. The Cancer Genome Atlas (TCGA) RNA-seq dataset (COADREAD) was obtained through cBioPortal (https://www.cbioportal.org/) and LinkedOmics (http://linkedomics.org/cptac-colon/), as previously described [3,15,31,32]. Four microarray datasets, GSE39582, GSE33113, GSE14333 and GSE37892, were downloaded from the Gene Expression Omnibus (GEO) repository (http://www.ncbi.nlm.nih.gov/geo), and a microarray dataset KFSYCC was downloaded from the CRCSC synapse data portal (Appendix A). In addition, we used an RNA-seq dataset (GSE171682) of CRC organoids and matched primary CRC tissues [33], and a microarray dataset (GSE100550) that profiled CRC patient tumors, patient-derived xenografts, patient-derived organoids and cell lines (Appendix A) [34].

The gene expression-based signatures used in this study were previously described [15]. Briefly, we built a TGFβ-stromal signature, designated TBSS, on the basis of a set of TGFβ-responsive stromal genes, including *VCAN*, *SERPINE1*, *CALD1*, *FAP, POSTN* and *IGFBP7* [35]. Tumors were dichotomized into TBSS^Low^ or TBSS^High^ based on the median. We also estimated the overall TGFβ levels based on the expression of *TGFB1* and *TGFB3*, designated *TGFB* [4]. The M2 macrophage signature was built based on a set of genes encoding M2 surface markers, including *CD163*, *MSR1* (CD204), *MRC1* (CD206), *CD200*, *FCER2* (CD23), *CD36*, *CD209* and *SLAMF1* (CD150) [15]. We also generated a set of genes assembled from published signatures, including Stromal ESTIMATE, which infers the faction of stromal cells, the C-ECM-UP signature, correlating with the dysregulation of ECM, and mesenchymal genes associated with epithelial-to-mesenchymal transition (EMT), designated stromal/ECM/mesenchymal genes [16,36,37].

### 2.2. Patient Samples

We enrolled forty-five consecutive patients with stage I–IV colorectal adenocarcinoma who underwent surgery at Fukushima Medical University Hospital between 2019 and 2021 without preoperative treatment, such as self-expanding metal stents, chemotherapy or radiotherapy. We used formalin-fixed paraffin-embedded (FFPE) whole tissue sections for immunohistochemistry. Tumors were classified according to the Japanese Classification of Colorectal, Appendiceal and Anal Carcinoma. The study was conducted in accordance with the Declaration of Helsinki and approved by the Institutional Review Board of Fukushima Medical University.

### 2.3. Immunohistochemistry

Four-μm thick FFPE sections were deparaffinized in xylene and rehydrated in ethanol. Endogenous peroxidases were blocked with 0.3% hydrogen peroxide in methanol. Antigens were retrieved by autoclave in 10 mM citrate buffer solution (105 °C, pH 6.0) for 5 min for VCAN, or by autoclave in Target Retrieval Solution (Agilent Technologies, Santa Clara, CA, USA) (100 °C, pH 9.0) for 10 min for TIM-3. Slides were incubated with primary rabbit polyclonal anti-VCAN antibody (HPA004726, Prestige Antibodies^®^ Powered by Atlas Antibodies, Sigma-Aldrich, St Louis, MO, USA; 1:500) or primary rabbit monoclonal anti-TIM-3 antibody (D5D5R XP^®^; #45208; Cell Signaling Technology, Danvers, MA, USA; 1:400) at 4 °C overnight, and then detected using a horseradish peroxidase-coupled anti-rabbit polymer (K4003; Envision+ system, Agilent Technologies). Peroxidase was visualized with diaminobenzidine (Dojindo, Kumamoto, Japan), and nuclei were counterstained with hematoxylin.

### 2.4. Assessment of Staining

Digital images were acquired using a digital slide scanner (NanoZoomer-SQ; Hamamatsu Photonics, Hamamatsu, Japan). The scanned images were independently evaluated by two observers (M. Katagata and H. Okayama), who had no prior knowledge of the clinical data. The assessment of VCAN staining was previously described [35]. Briefly, the percentage positivity of stromal VCAN was scored as 0 (0–5%), 1 (5–25%) or 2 (>25%) in the tumor stromal area, and the intensity of staining was scored as 0 (negative), 1 (moderate) or 2 (strong). The scores were then combined to generate a VCAN score (min 0, max 4). For TIM-3 staining, we evaluated five independent high-power microscopic fields at the hotspot areas of each tumor. The proportion of TIM-3-stained tumor-infiltrating immune cells (TIICs) was graded as 0 (<1%), 1 (1–5%), 2 (5–20%), 3 (20–50%) or 4 (>50%), and the intensity of staining was graded as 0 (negative), 1 (weak), 2 (moderate) or 3 (strong). We then multiplied the scores to obtain the total staining score for one field (min 0, max 12), and the final TIM-3^+^ TIICs score for each tumor was the mean of the five fields. We further selected tumors with a VCAN score of 3 or 4 that showed a heterogeneous VCAN staining pattern, and then evaluated the TIM-3^+^ TIICs in stromal areas with strong VCAN staining (VCAN^High^ areas) and those with negative or moderate VCAN staining (VCAN^Low^ areas).

### 2.5. Immunofluorescence

The FFPE sections were deparaffinized and rehydrated. Antigens were retrieved by autoclave for 20 min in Target Retrieval Solution (Agilent Technologies) (100 °C, pH 9.0). Blocking was performed with 2% donkey serum in TBS/T for 30 min. Thereafter, the sections were stained with anti-human CD163 antibody (NCL-L-CD163; Leica biosystems, Wetzlar, Germany; 1:200) and anti-human TIM-3 antibody (D5D5R XP^®^; #45208; Cell Signaling Technology; 1:200) at 4 °C overnight. The sections were incubated for 1 h with Alexa Fluor 488-conjugated anti-mouse (A-21202; Thermo Fisher Scientific, Waltham, MA, USA; 1:200) and Alexa Fluor 555-conjugated anti-rabbit (A-31572; Thermo Fisher Scientific; 1:200) secondary antibodies. Nuclei were stained with DAPI (D9542; Sigma-Aldrich; 40 ng/mL) for 10 min. Finally, the slides were mounted with ProLong Glass Antifade Mountant (P36984; Thermo Fisher Scientific), and images were obtained using an Olympus FV1000-D confocal microscope.

### 2.6. Flow Cytometry Analysis

Cells were stained with antibodies, including APC/Cyanine7-conjugated anti-human CD14 mAb (325620, BioLegend, San Diego, CA, USA), PE/Cyanine7-conjugated anti-human CD163 mAb (333613, BioLegend), Pacific Blue-conjugated anti-human CD86 mAb (305423, BioLegend), 7-AAD Viability Staining Solution (420404, BioLegend), PE-conjugated anti-human TIM-3 mAb (345005, BioLegend) and PE-conjugated isotype control mAb (400111, BioLegend). Unstained samples were used as the negative controls. The stained cells were measured using a BD FACS Canto II flow cytometer (BD Biosciences, San Jose, CA, USA) and the data were analyzed using FlowJo software (version 10.8.1, FlowJo, Ashland, OR, USA).

### 2.7. Generation of M2 and M1 Macrophages

M2 and M1 macrophages were generated by the generally established methods [38,39]. Human peripheral blood mononuclear cells (PBMCs) were collected from the peripheral blood of healthy volunteers using a BD Vacutainer^®^ CPT™ (BD Bioscience), and monocytes were isolated using a MidiMACS™ Separator (130-042-302, Miltenybiotec, Bergisch Gladbach, Germany), LS column (130-042-401, Miltenybiotec) and CD14 MicroBeads human (130-050-201, Miltenybiotec). CD14^+^ monocytes were cultured in a 12-well plate at a density of 5 × 10^5^ per well at 37 °C in 5% CO_2_ atmosphere in RPMI1640 (Sigma-Aldrich) supplemented with 20 ng/mL of M-CSF (216-MC, R&D systems, Minneapolis, MN, USA), penicillin/streptomycin and 10% heat-inactivated FBS for 6 days. Subsequently, we continued to culture the cells in the same medium for 3 days to generate M0 macrophages, while we added 20 ng/mL each of IL-4 (204-IL, R&D Systems), IL-10 (217-IL, R&D systems) and IL-13 (213-ILB, R&D systems) into the culture medium and continued to culture the cells for 3 days for the generation of M2 macrophages. To differentiate M1 macrophages, we utilized 20 ng/mL of GM-CSF (7954-GM, R&D Systems) and 20 ng/mL each of LPS (L2630, Sigma-Aldrich), IFN-γ (285-IF, R&D Systems) and IL-6 (206-IL, R&D Systems).

### 2.8. TGFβ Treatment

PBMCs were cultured in a 12-well plate at a density of 5 × 10^5^ per well at 37 °C in 5% CO_2_ atmosphere in RPMI1640 (Sigma-Aldrich) supplemented with 10% heat-inactivated FBS and penicillin/streptomycin, and were exposed to 0, 0.1, 1 or 10 ng/mL of TGFβ1 (7754-BH, R&D systems). Cells were harvested 48 h after treatment initiation and used in each experiment. To generate IL-2-stimulated T-cells, PBMCs were stimulated with complete medium alone or with human IL-2 (589102, BioLgend) at 25 ng/mL for 6 days, and the cells were then treated with 10 ng/mL of TGFβ1 in the presence of IL-2 for 48 h.

### 2.9. Statistical Analysis

Correlations were assessed using Spearman’s coefficient. The Mann–Whitney *U* test or paired Wilcoxon signed-rank test was used to determine differences between two variables. The Kruskal-Wallis test with Dunn’s post hoc test was used to compare multiple groups. All statistical analyses were conducted using GraphPad Prism v9.5.1 (GraphPad Software Inc., San Diego, CA, USA). All *p*-values were two-sided, and we considered *p*-values less than 0.05 to be statistically significant.

## 3. Results

### 3.1. Correlation between the Expression of HAVCR2, M2 Macrophage and TBSS in MSI and MSS CRC

In our recent study, which focused on MSI CRC, we developed the TGFβ-stromal signature (TBSS), by which to infer the levels of TGFβ-dependent stromal activation, based on six TGFβ-responsive, stroma-specific genes, consisting of *VCAN*, *POSTN*, *FAP*, *CALD1*, *SERPINE1* and *IGFBP7*, and identified a CMS4-like, TGFβ-activated stromal subset within MSI CRCs characterized by high levels of M2 macrophage signature, ICI resistance signature and poor prognosis [15]. Notably, this stromal MSI subtype also consistently displayed the upregulation of *HAVCR2* (encoding TIM-3), but not of the other immune checkpoint genes.

In the present study, we first examined whether the association of the TGFβ-rich TME with *HAVCR2* expression and M2 macrophage signature is characteristic of MSI tumors or irrespective of MSI status. Using the TCGA RNA-seq dataset, although MSI tumors showed higher levels of *HAVCR2* expression and M2 macrophage signature than those of MSS, we found that the levels of *HAVCR2* expression, M2 macrophage signature and TBSS were significantly correlated with *TGFB* levels in the analysis of both MSI and MSS tumors (Figure 1A–C and Appendix A). Therefore, we decided to analyze MSI and MSS CRCs together. We differentially examined expressed genes between TBSS^Low^ and TBSS^High^ tumors and found the striking upregulation of *HAVCR2* (log_2_ fold change = 1.80, *p* = 2.76 × 10^−46^), along with marked enrichment of *TGFB* genes, stromal/mesenchymal/ECM genes and M2 macrophage marker genes, such as *MSR1* (CD204), *CD163* (CD163) and *MRC1* (CD206), in TBSS^High^ tumors (Figure 1D). It is worth noting that the expression of *HAVCR2* had a much closer relationship with high TBSS, among the other co-inhibitory receptor genes, including *TIGIT*, *CTLA4*, *PDCD1* (PD-1) and *LAG3*, prompting us to further explore the role of *HAVCR2* in the TGFβ-rich TME in CRC.

### 3.2. The Expression of HAVCR2 and M2 Macrophage Signature in the TGFβ-Activated Stroma in CMS1 and CMS4 CRC

To validate the strong association of *TGFB* with *HAVCR2*, M2 macrophage and TBSS, which was found to be independent of MSI status, we utilized multiple independent cohorts (*n* = 1975, in total), including TCGA RNA-seq and five microarray datasets in conjunction with CMS classification and MSI status (Appendix A). In all six cohorts, we confirmed significant positive correlations of *TGFB* levels with *HAVCR2*, M2 macrophage signature and TBSS (Figure 1E). These variables were also significantly correlated with one another in each of the cohorts (Appendix A). These data consistently indicated that the expression of *HAVCR2* along with M2 macrophage signature, was strongly upregulated in the TGFβ-activated TME. When focusing on the CMS subtypes and MSI status, *TGFB*, *HAVCR2*, M2 macrophage and TBSS were together found to be significantly higher in both CMS1 (mainly MSI) and CMS4 tumors, compared to those of CMS2 and CMS3 in all cohorts (Figure 2). Collectively, although *TGFB*, *HAVCR2*, M2 macrophage and TBSS were robustly correlated with each other regardless of MSI status or CMS subtype, they were found to be strongly co-upregulated in MSI, CMS1 and CMS4 CRCs.

The transcriptomic data from bulk tumor samples (Figure 1 and Figure 2) consisted of mixed signals originating from both epithelial tumor cells and many types of stromal cells in the TME, in which most cells have the potential to respond to TGFβ, and the responses largely depend on cell types [8]. Therefore, we sought to determine whether the *TGFB*-dependency of *HAVCR2*, M2 macrophage and TBSS found in patient cohorts were stroma-specific and unrelated to the signals derived from the epithelial component. To this end, we further analyzed transcriptomic datasets of CRC bulk tissues, patient-derived organoids, patient-derived xenografts and cell lines. Organoids and cell lines lack a stromal component, whereas, in xenografts, cancer stroma is replaced by murine counterparts and is not or only partially detected by human-specific expression platforms. Based on pairwise analysis of patient-derived CRC organoids and their matched CRC bulk tissues (GSE171862), we found that the levels of *TGFB* were significantly correlated with *HAVCR2*, M2 macrophage and TBSS in bulk tumor tissues (Spearman’s *r*: 0.6716–0.8463, *p* < 0.0001) but not in organoids (Spearman’s *r*: −0.01699–0.3112) (Figure 3A–C). Correspondingly, in GSE100550, a significant positive correlation of *TGFB* with *HAVCR2*, M2 macrophage and TBSS was specifically found in bulk CRC tissues (Spearman’s *r*: 0.5792–0.7909, *p* < 0.0001), but not in CRC organoids, xenografts or cell lines (Spearman’s *r*: −0.2239–0.3333) (Figure 3D–F). These data clearly support the notion that the *TGFB*-dependent increase in *HAVCR2*, M2 macrophage and TBSS in CRC exclusively arises from the cancer stroma.

### 3.3. TIM-3^+^ Tumor-Infiltrating Immune Cells in TGFβ-Activated Stroma

We next conducted immunohistochemistry for TIM-3 and VCAN, a TGFβ-responsive stromal protein also known as versican, using surgically resected CRC tissues. Stromal VCAN immunostaining can indicate TGFβ-activated stroma, as it was exclusively stained in the cancer stroma, mainly expressed by CAFs in association with phospho-SMAD3 staining [15,35]. VCAN IHC scores and TIM-3^+^ tumor-infiltrating immune cells (TIICs) were evaluated using 45 CRC specimens. Representative images of VCAN and TIM-3 are shown in Figure 4A–F and Appendix A. Overall, TIM-3^+^ TIICs were often found in the tumor center, tumor invasive front and lymphoid structures in the TME, and TIM-3 was mainly expressed by macrophage-like TIICs (Appendix A). Moreover, immunofluorescent staining for TIM-3 and an M2 macrophage marker, CD163, in CRC tissues confirmed that TIM-3 exhibited a high degree of overlap with CD163^+^ macrophages (Figure 4G). We found that tumors with higher stromal VCAN scores were significantly associated with higher levels of TIM-3^+^ TIICs in the TME (Figure 4H, Spearman’s *r* = 0.584, *p* < 0.0001). Considering the heterogeneous patterns of VCAN and TIM-3 staining within tumors, we used 24 tumors showing heterogeneous VCAN staining, and the stromal areas of each tumor were further divided into those with strong VCAN staining (VCAN^High^ areas) and those with negative or moderate VCAN staining (VCAN^Low^ areas) (Appendix A). We found significantly higher levels of TIM-3^+^ TIICs in VCAN^High^ stromal areas than in VCAN^Low^ areas (Figure 4I, *p* < 0.0001).

### 3.4. M2 Macrophages Express TIM-3 and TGFβ Induces TIM-3 on Monocytes

We demonstrated that the close relationship between M2 macrophage signature and *TGFB* is consistent with the fact that TGFβ can contribute to inducing the M2-like macrophage phenotype, and that M2-like macrophages can be a major source of TGFβ in the TME [8,10,40,41]. Given the strong correlation between *HAVCR2* expression and M2 macrophage signature at the transcriptional level, we investigated the surface expression of TIM-3 in M2-polarized macrophages. Human CD14^+^ monocytes, isolated from PBMCs, were differentiated into the M2 phenotype in vitro, showing high levels of cell surface TIM-3 together with an M2 marker, CD163, compared to M0 macrophages by flow cytometry (Figure 5A,B and Appendix A). Although M1 macrophages were also generated from PBMCs, cell surface TIM-3 expression did not significantly differ between generated M0 and M1 macrophages (Figure 5C,D and Appendix A).

Since we found the *TGFB*-dependent increase in *HAVCR2* and M2 macrophage signature in transcriptomic datasets and dense infiltration of TIM-3^+^ TIICs in VCAN^+^ TGFβ-activated stroma by immunohistochemistry, we hypothesized that TGFβ may induce TIM-3 expression in immune cells, particularly in myeloid cells. To this end, we analyzed the expression of TIM-3 in monocytes, CD4^+^ T-cells and CD8^+^ T-cells obtained from PBMCs treated with varying concentrations of TGFβ in culture (Appendix A). We found that TGFβ induced a significant dose-dependent increase in TIM-3 expression in monocytes (Figure 5E,F), but not in CD4^+^ T-cells or CD8^+^ T-cells (Appendix A). Although we further utilized IL-2-stimulated T-cells expressing TIM-3, treatment with TGFβ did not significantly alter the levels of TIM-3 on the surface of CD4^+^ or CD8^+^ T-cells regardless of IL-2 stimulation (Appendix A).

## 4. Discussion

In the present study, we found a strong correlation between the TGFβ-dependent active TME and *HAVCR2* expression in CRC, and showed consistent results across multiple datasets, which consisted of approximately 2200 CRC tissues, organoids and xenografts. This study revealed that *HAVCR2* was strikingly upregulated in a considerable fraction of CMS4 and CMS1/MSI CRCs, both of which were characterized by the TGFβ-activated TME. These correlations at transcriptional levels were further recapitulated by immunohistochemistry. We found that the levels of tumor-infiltrating TIM-3^+^ immune cells, mainly M2-like macrophages, were correlated with tumors exhibiting TGFβ-activated stroma, as detected by VCAN staining, and that TIM-3^+^ cells particularly accumulated in VCAN^High^ stromal areas. We also demonstrated a significant correlation of *TGFB* with the expression of *HAVCR2* and M2 macrophage signature, and the upregulation of TIM-3 was observed on the surface of M2-polarized macrophages and TGFβ-stimulated monocytes. Therefore, in the TGFβ-rich TME, TIM-3 may be mainly expressed by M2-like macrophages at least in part induced by TGFβ. To our knowledge, this is the first study to address TIM-3 and M2 macrophages in the TGFβ-rich TME, in association with MSI status and CMS subtypes in CRC. Our results are in good agreement with a previous study on hepatocellular carcinoma by Yan et al., which reported that TGFβ not only induced TIM-3 on macrophages but also facilitated M2-like polarization in cultured macrophages [27]. Yan et al. also identified a SMAD-binding site in the 5′ region of TIM-3, and demonstrated that TGFβ enhanced the TIM-3 promotor activity examined by luciferase assays [27]. TIM-3 can promote M2 macrophage polarization through several mechanisms in CRC and HCC [21,27,29]. Collectively, several lines of evidence indicate a mechanistic link between the M2-like polarization of macrophages and increased TIM-3 expression in the TGFβ-rich TME. It is also worth noting that TIM-3, M2-like macrophages and the TGFβ-activated TME are known to have tumor-promoting and immunosuppressive functions, as well as poor prognostic impact in solid cancers, particularly in CRC [4,5,6,7,8,24,30,42].

TGFβ is a pleiotropic cytokine with complex and context-dependent functions in cancer. TGFβ is known to be a potent growth inhibitor of cells of epithelial origin, thus, it acts as a tumor-suppressor in pre-malignant cells in the early stages of tumorigenesis [10]. During CRC progression, TGFβ levels increase in the TME [6]. Although it is well documented that TGFβ promotes cancer cell-intrinsic effects on invasion and metastasis associated with EMT, cancer cells often acquire loss-of-function mutations in TGFβ signaling pathway genes to avoid the growth inhibitory effect of TGFβ [6,10]. Consequently, in advanced-stage tumors, TGFβ plays a key role in shaping the tumor-promoting TME and driving immune evasion, where CAFs are activated and TAMs are polarized toward the M2 phenotype with both cell types being the major sources of TGFβ [8,10,40,41]. TGFβ has also emerged as a negative predictor of ICI response. In clinical trials and animal models, TGFβ-activated CAFs and TGFβ-induced ECM genes are associated with a poor response to anti-PD-1/PD-L1 therapies in several solid cancers [8,16,17,18,19]. Moreover, several strategies for the dual inhibition of TGFβ and PD-1/PD-L1 signaling have been extensively studied in preclinical models, showing uniformly positive results across multiple types of solid cancers [17,20,43,44,45]. In mouse models of MSS CRC with CMS4 phenotype, TGFβ inhibition exerted synergic anti-tumor effects with anti-PD-L1 therapy [9], while we previously identified a CMS4-like subgroup within MSI CRCs displaying TGFβ-active stroma as well as an ICI resistance signature [15]. Therefore, the combined blockade of TGFβ and PD-1/PD-L1 may provide clinically meaningful responses in some patients with MSI and MSS CRC, possibly by making the immunosuppressive TME sensitive to ICIs. However, despite the success in preclinical models, clinical trials of anti-TGFβ combined with ICIs in several solid cancers, including CRC, have suggested that this approach may be limited, as the addition of TGFβ inhibitors often failed to show meaningful clinical benefits beyond ICIs alone in unselected patients [20]. In addition, the systemic adverse effects of inhibiting TGFβ signaling and its narrow therapeutic window require careful consideration [8,20]. Although some trials have already been terminated and many others are currently underway, it may be suggested that patient selection and additional therapeutic combinations are required to maximize the benefit of these strategies. For example, CMS4 CRCs (mostly MSS), as well as a CMS4-like subset of MSI and/or CMS1 CRCs, might be a candidate population that may be effectively treated with anti-TGFβ combined with anti-PD-1/PD-L1 therapies, as they have a worse prognosis and exhibit much higher levels of TGFβ-dependent stromal activation than those of CMS2 and CMS3. More importantly, TIM-3 can be an actionable target in combination with a PD-1/PD-L1 blockade for these patients. The dual inhibition of PD-1 and TIM-3 showed synergic effects in animal models [22,46], and adaptive resistance to the PD-1 blockade was associated with the upregulation of TIM-3 [23]. Anti-TIM-3 antibodies, mostly in combination with an anti-PD-1/PD-L1 blockade, are currently being investigated in clinical trials for the treatment of advanced cancers, and have shown encouraging early efficacy results with no serious adverse events or dose-limiting toxicities [21,30,47]. Concerning the adverse events observed in clinical trials of TGFβ inhibition [8,20], the blockade of TIM-3 may be a safe and promising alternative, at least in selected patients with CRC. Therefore, we suggest that, in future trials, dual TIM-3 and PD-1/PD-L1 inhibition should be considered for patients with CMS4 and MSI CRC, where TIM-3 is highly expressed in the TGFβ-rich TME.

## 5. Conclusions

In conclusion, using large-scale transcriptomic data analysis, followed by immunohistochemistry and in vitro experiments, we demonstrated a mechanistic link between TIM-3 expression and M2-like macrophages in the TGFβ-activated TME. We found that *TGFB*, *HAVCR2*, M2 macrophage and TBSS signatures were specifically derived from the stromal components and robustly correlated with each other, and that they were co-upregulated in CMS4, CMS1 and MSI CRCs. Since TGFβ and TIM-3 are implicated in immune evasion and poor response to PD-1/PD-L1 blockade, immunotherapeutic strategies targeting TIM-3 and/or TGFβ combined with PD-1/PD-L1 signaling may overcome the resistance to current ICIs, particularly for CMS4 and MSI CRCs characterized by high levels of TIM-3 and TGFβ.

## Figures and Tables

**Figure 1 cancers-15-04943-f001:**
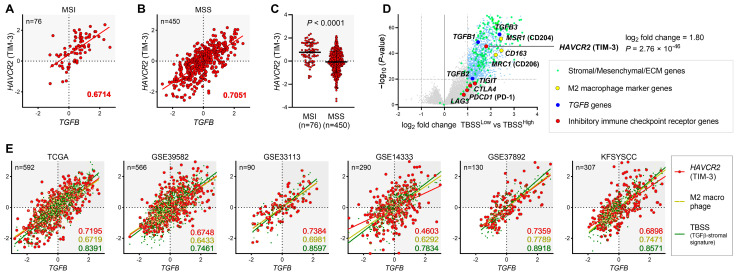
Association of *TGFB* with *HAVCR2*, M2 macrophage signature and TGFβ-stromal signature (TBSS) in multiple RNA-seq or microarray cohorts of CRC in conjunction with MSI status and CMS subtypes. (**A**,**B**) Correlations between the expression of *TGFB* and *HAVCR2* in MSI (**A**) and MSS (**B**) CRC. Spearman’s coefficients are indicated (*p* < 0.0001). (**C**) *HAVCR2* expression in MSI and MSS CRC. (**D**) Volcano plot displaying differentially expressed genes between TBSS^Low^ and TBSS^High^ tumors. The x-axis shows the log_2_-fold change, and the y-axis displays the -log_10_ of the *p*-values. (**E**) Correlations of *TGFB* with *HAVCR2* (red), M2 macrophage (yellow) and TBSS (green) in six independent cohorts of CRC. Spearman’s coefficients are shown as colors corresponding to *HAVCR2* (red), M2 macrophage (yellow) and TBSS (green) for each dataset. *p*-values for all correlations < 0.0001.

**Figure 2 cancers-15-04943-f002:**
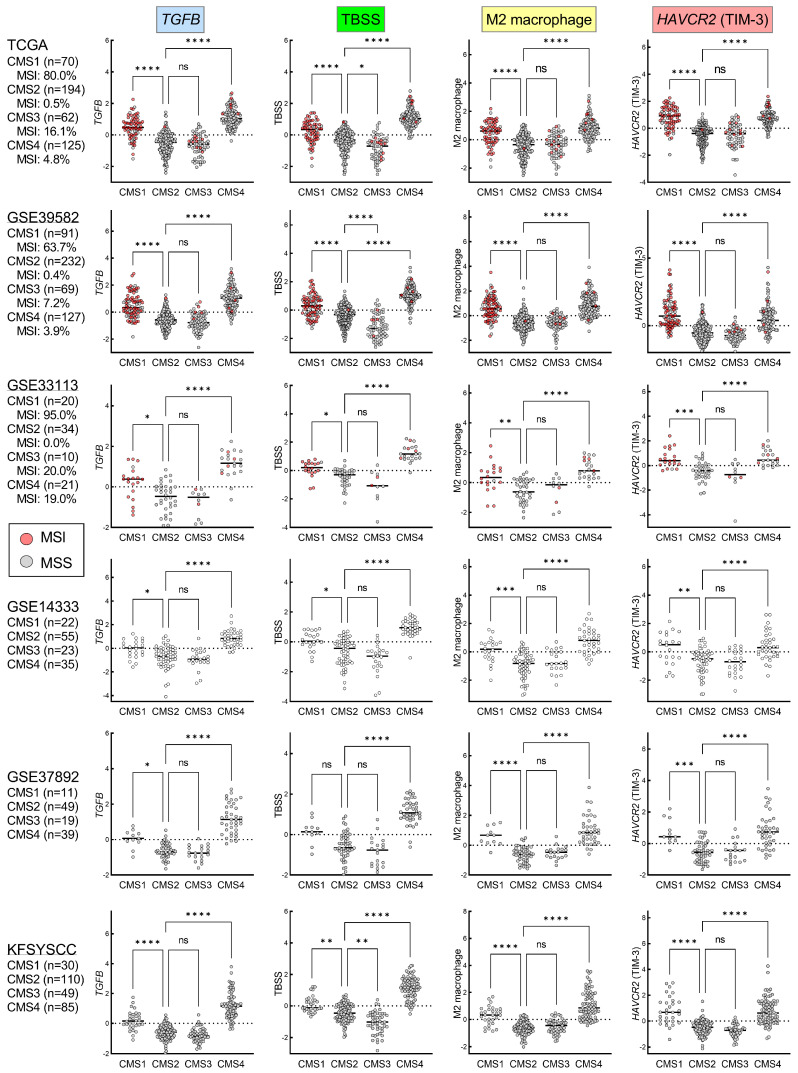
Association between CMS subtypes and the levels of *TGFB*, TGFβ-stromal signature (TBSS), M2 macrophage signature and *HAVCR2* expression in six independent cohorts of CRC, including TCGA, GSE39582, GSE33113, GSE14333, GSE37892 and KFSYSCC. MSI status was available for the former three cohorts; red dots represent MSI tumors. Kruskal-Wallis test, followed by Dunn’s post hoc test to compare each CMS subtype with CMS2. **** *p* < 0.0001, *** *p* < 0.001, ** *p* < 0.01, * *p* < 0.05, ns *p* > 0.05.

**Figure 3 cancers-15-04943-f003:**
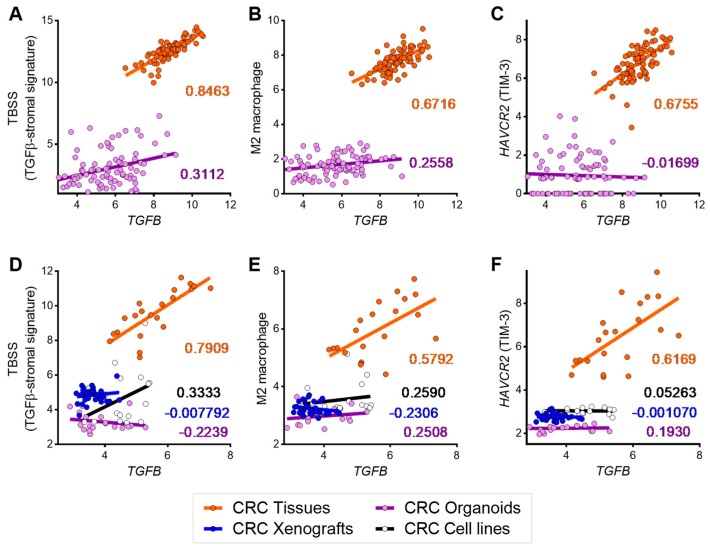
Stroma-specific association of *TGFB* with TGFβ-stromal signature (TBSS), M2 macrophage signature and *HAVCR2*. (**A**–**C**) Primary CRC tissues (orange) and matched organoids (purple) in an RNA-seq dataset (GSE171682). (**D**–**F**) CRC tissues (orange), organoids (purple), xenografts (blue) and cell lines (black) in a microarray dataset (GSE100550). Spearman’s coefficients are indicated as colored numbers corresponding to tissues (orange), organoids (purple), xenografts (blue) and cell lines (black) for each dataset.

**Figure 4 cancers-15-04943-f004:**
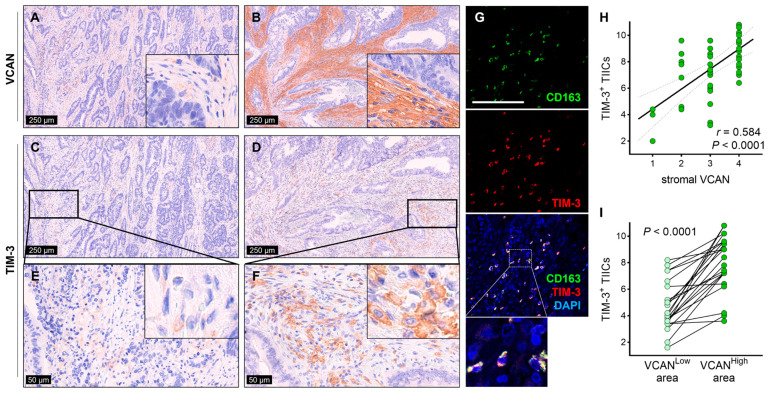
Immunohistochemistry for a TGFβ-responsive stromal protein, VCAN (**A**,**B**), and TIM-3^+^ tumor-infiltrating immune cells (TIICs) (**C**–**F**) in CRC tissues. (**A**,**C**,**E**) Serial sections from a representative case of CRC showing negative stromal VCAN staining (**A**) with low infiltration of TIM-3^+^ TIICs (**C**,**E**). (**B**,**D**,**F**) Serial sections from a representative case of CRC showing strong stromal VCAN staining (**B**) with dense TIM-3^+^ TIICs (**D**,**F**). (**G**) Representative immunofluorescence of CD163 (green), TIM-3 (red) and DAPI (blue) in CRC tissues. Scale bar, 100μm. (**H**) Association between the levels of TIM-3^+^ TIICs and stromal VCAN in CRC (*n* = 45) by Spearman’s correlation. (**I**) Association between TIM-3^+^ TIICs and heterogenous VCAN expression patterns (VCAN^Low^ areas and VCAN^High^ areas) within the tumor (*n* = 24). Pairwise significance was tested using the Wilcoxon signed rank test.

**Figure 5 cancers-15-04943-f005:**
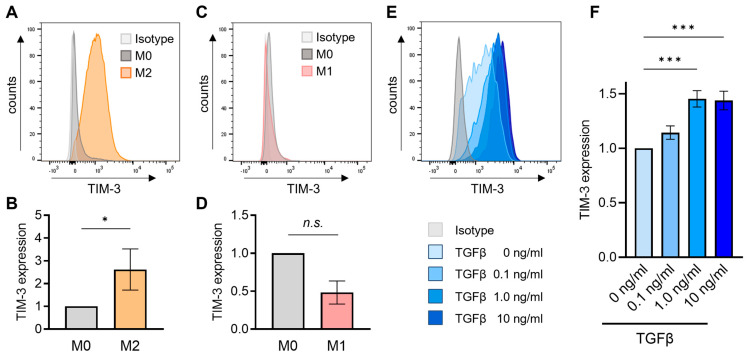
TIM-3 expression on M2 and M1-polarized macrophages and TGFβ-treated monocytes. (**A**,**B**) TIM-3 expression on M0 and M2-polarized macrophages. Representative histograms for TIM-3 are shown (**A**), and the expression levels of TIM-3 on M2 macrophages present as the mean ± s.e.m. relative to those of M0 macrophages (*n* = 4) (**B**). Mann–Whitney *U* test. * *p* < 0.05. (**C**,**D**) TIM-3 expression on M0 and M1-polarized macrophages. Representative histograms for TIM-3 (**C**), and the levels of TIM-3 on M1 macrophages as the mean ± s.e.m. relative to those of M0 macrophages (*n* = 3) (**D**). *n.s. p* > 0.05. (**E**,**F**) Representative histograms displaying TIM-3 expression on monocytes treated with various concentrations of TGFβ (**E**), and TIM-3 expression levels on TGFβ-treated monocytes present as the mean ± s.e.m. relative to those of controls (*n* = 10) (**F**). Kruskal-Wallis test with Dunn’s post hoc test. *** *p* < 0.001.

## Data Availability

The public datasets used in this study are available as described earlier. The data generated in this study are available from the corresponding author upon reasonable request.

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
