# Peer review of "TIM-3 Expression and M2 Polarization of Macrophages in the TGFβ-Activated Tumor Microenvironment in Colorectal Cancer"

_cancers, 2023, doi:10.3390/cancers15204943_

Round 1
Reviewer 1 Report
Katagata et al, present an interesting analysis in colorectal cancer samples based mainly in RNA public data, but also supported by data from cell lines, patient-derived organoids, CRC xenograft and CRC tissues, immunohistochemistry and a small functional analysis. They analyzed stromal markers (TBSS a TGFB stromal signature and TGFB), an exhausted marker (HAVCR2) and a specific marker of macrophage polarization (M2). Basically, the authors show a good correlation between these markers (TBSS, TGFB, HAVCR2 and M2) and an enrichment of these markers in CMS1 and CMS4 subtypes. Interestingly these markers are enriched in both MSI and MSS CMS1 and CMS4 subtypes and poorly enriched in CMS2 and CMS3 subtypes.
Major comments:
- Patients with MSI subtype are specially enriched in CMS1 subtype, followed by CMS3 and CMS4 subtype. Patients with CMS2 subtype (the more frequent in CRC) are rarely MSI. In Fig S3 it seems that the majority of MSI patients not only fulfil CMS1 subtype, but also seems to over-express the markers (TBSS, TGFB, HAVCR2 and M2) more than in MSS patients. I suggest that the numbers and proportions will be settle in the manuscript emphasizing the proportion of MSI vs MSS patients that express these markers (TBSS, TGFB, HAVCR2 and M2) in each of CMS subtypes. I suggest also to replace Fig 1E and F for the figure S3B
- Because most of the MSI patients show high expression of TBSS, TGFB, HAVCR2 and M2 the authors should explain in the discussion, why these patients (MSI) are still responding to immune check-point inhibitors (ICB). This is particularly important because despite pre-clinical data, anti-TGFB inhibitors plus ICB failed in patients with MSS CRC to improve efficacy. Therefore, it seems more a patient tumor characteristic than a biomarker of ICB efficacy. I suggest to improve the discussion based on the results specially those that suggest that TIM-3 and TGFB inhibitor can overpass ICB resistance.
Minor comments:
- In page 2 line 66 I suggest to write dysfunctional exhausted instead of dysfunctional and exhausted T-cell….
- In page 7 line 293 I suggest to stop the phrase in ..and cell lines. Organoids and cell lines….
- I suggest to avoid all the phrases with MSI/CMS1 and MSS/CMS4
Reviewer 2 Report
This article focuses on the relationship between TGFβ, TIM-3, and immune response in colorectal cancer (CRC). The study uses large-scale transcriptomic data analysis, immunohistochemistry, and in vitro experiments to reveal a connection between TIM-3 expression and the presence of M2-like polarized macrophages in the TGFβ-rich tumor microenvironment (TME) of CRC. These findings suggest that TIM-3 and TGFβ play crucial roles in immune evasion and resistance to anti-PD-1/PD-L1 immune checkpoint inhibitors (ICIs) in CRC, particularly in CMS4 and MSI/CMS1 CRCs where TIM-3 and TGFβ are highly expressed. The study proposes that targeting TGFβ or TIM-3 in combination with ICIs could be a promising strategy to overcome ICI resistance in these specific CRC subtypes. Additionally, it highlights the potential of TIM-3 as a target for immunotherapy in CRC and the need for patient selection to maximize the benefits of these strategies.
The article's focus on the interplay between TGFβ, TIM-3, and the immune response within the CRC tumor microenvironment (TME) is particularly timely with TIM-3 as a major immunotherapeutic target. The authors’ comprehensive approach, including analysis of large-scale transcriptomic datasets, immunohistochemistry, and in vitro experiments, helps paint an interesting story for the reader and lends credibility to the findings. The methodology is robust, the results are compelling, and the implications for potential therapies are promising. I believe that this article is suitable for publication in Cancers, where it can contribute significantly to the field of CRC research and immunotherapy. The authors present a well-executed study with valuable insights into the relationship between TGFβ, TIM-3, and the immune response in colorectal cancer.
Round 2
Reviewer 1 Report
I agree with the changes in the manuscript